# Clarification of fibrin generation and degradation reaction processes of clot-fibrinolysis waveform in hemorrhagic disorders

Tomoko Matsumoto[1]*, Nukumi Tujii[2,3], Daiki Shimomura[3], Aya Kouno[3], Takeshi Suzuki[4], Sho Shinohara[4], Nobuo Arai[4], Hiroshi Kurono[4], Osamu Kumano[4,5], Mikio Kamioka[3]

1 Department of Clinical Laboratory Science, Tenri University, Tenri City, Nara, Japan, 2 Department of Clinical Laboratory Science, Tenri Health Care University, Tenri City, Nara, Japan, 3 Department of Laboratory Medicine, Tenri Hospital, Tenri City, Nara, Japan, 4 Sysmex Corporation, Kobe, Japan, 5 Health and Medical Research Institute, National Institute of Advanced Industrial Science and Technology (AIST), Takamatsu, Japan

* t.matsumoto@sta.tenri-u.ac.jp

## Abstract

Clot-fibrinolysis waveform analysis (CFWA) is an assay used to simultaneously evaluate coagulation and fibrinolysis reactions. Although the assay detected the reaction via transmittance changes, there was no evidence that the transmittance changes indicated reactions. This study aimed to demonstrate that transmittance changes indicate coagulation and fibrinolysis reactions by detecting relative markers. CFWA was conducted using activated partial thromboplastin time (APTT) reagent and $CaCl_2$ solution with tissue-plasminogen activator (tPA); transmittance changes were monitored, and the first derivative curve was analyzed in pooled normal plasma (PNP) and factors V, VIII, IX, X, and XI-deficient plasma samples. The samples during the coagulation and fibrinolysis reactions were prepared by adding the reaction stop solution, fibrin monomer complex (FMC), fibrin/fibrinogen degradation products (FDP), D-dimer and plasmin-$\alpha_2$ plasmin inhibitor complex (PAP) were measured to compare the waveform with the tendencies of these markers. The fibrinolysis markers FDP, D-dimer, and PAP increased in all samples as the reaction time increased. In FMC, the value increased during the coagulation phase, decreased at the end of the phase, and increased again during the fibrinolysis phase. FMC, FDP, and D-dimers were generated from fibrin/fibrinogen in the CFWA assay, indicating that the assay reflects coagulation and fibrinolysis reactions by monitoring transmittance.

## Introduction

Various assays have been widely used to estimate blood coagulation and fibrinolysis status in patients. Global assays, including activated partial thromboplastin time (APTT) and prothrombin time (PT), are useful screening tests. Clot waveform

**Data availability statement:** All relevant data are within the manuscript and its Supporting Information files.

**Funding:** This study was supported by JSPS KAKENHI (grant number: JP 21K09089).

**Competing interests:** The first author has no conflicts of interest. T. Suzuki, S. Shinohara, N. Arai, and H. Kurono are employees of Sysmex Corporation. O. Kumano was an employee of the Sysmex Corporation when the study was conducted.

analysis (CWA) is an excellent and innovative method for the quantitative evaluation of coagulation reactions [1]. Light changes were monitored in the analyzer during the assay, and the measurement data were described as a clot waveform. This waveform includes various types of information and expands the interpretation of the measurement results. CWA using an APTT reagent has been widely employed and its usefulness has been reported in numerous studies [2–5]. Currently, CWA are widely recognized for their usefulness, and several application methods have been reported.

Recently, clot-fibrinolysis waveform analysis (CFWA) was developed as an assay to evaluate both coagulation and fibrinolysis reactions simultaneously using APTT reagent and $CaCl_2$ solution with tissue plasminogen activator (tPA) [6]. The thromboelastography can evaluate coagulation and fibrinolysis, but it has low reproducibility [7]. The clot lysis time (CLT) with thrombin addition can assess fibrinolysis, but it does not reflect intrinsic or extrinsic coagulation [8]. CFWA uses plasma as the measurement sample and an automated coagulation analyser, so results are available within 10 minutes and it has excellent reproducibility, and it can use plasma stored in a freezer. This investigation was performed using a modified APTT reagent to evaluate the bleeding tendency caused by a decrease in clotting factors, including hemophilia, which is an endogenous coagulation disorder. CFWA can assess not only the coagulation phase but also the fibrinolysis process. In the detailed mechanism: i) the transmittance decreases in the first reaction, ii) the transmittance increases in the second reaction after the transmittance in the first reaction plateaus, and iii) the transmittance in the second reaction reaches a level to the starting point in the first reaction (Fig 1a). The first and second reactions are coagulation and fibrinolysis, respectively. CFWA has been used as a unique tool to analyze coagulation and fibrinolysis, and the assay can assess the direct oral anticoagulant (DOAC) *in vitro* effects and investigate the hemostatic balance in the COVID-19 situation [9–11]. The basic CFWA characteristics were investigated in coagulation factor-deficient and drug-spiked samples [6,12–15]. Furthermore, the relationship between CFWA and fibrinolysis markers has been investigated and it has recently been shown that CFWA is related to the fibrinogen, fibrin monomer complex (FMC), $\alpha_2$-plasmin inhibitor, and plasminogen concentrations. It was suggested that the assay has the potential to reflect the fibrinolysis status in one global assay [16].

However, CFWA does not directly detect the enzymatic activity of coagulation and fibrinolysis factors, and there is no evidence that transmittance changes directly indicate these reactions. In addition, it has not been shown that fibrin degradation products are generated from the clot in the coagulation and fibrinolysis reactions during the assay. Therefore, we investigated whether transmittance changes reflected coagulation and fibrinolysis reactions by detecting the proteins generated during these reactions using clinical laboratory testing and immunoblotting. This study aimed to demonstrate that changes in transmittance indicate coagulation and fibrinolysis reactions, using protein detection. Furthermore, we compared the transmittance changes in normal plasma samples with those in coagulation factor-deficient plasma samples.

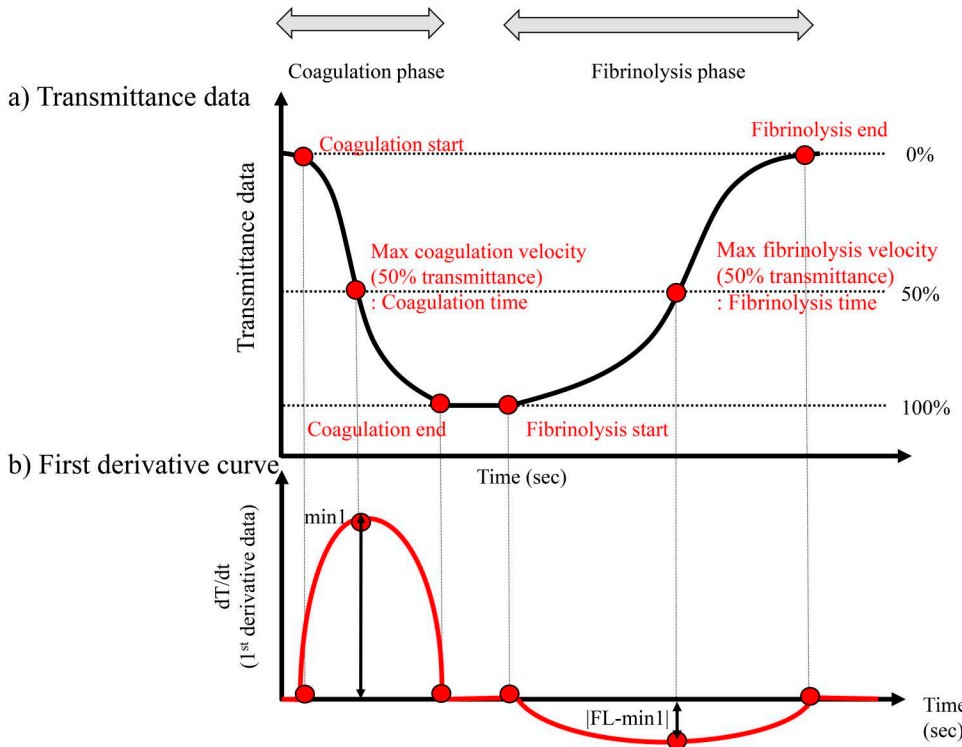

**Fig 1. Overview of clot-fibrinolysis waveform analysis point of time and parameters. a)** Transmittance data was monitored in the analyzer, and the typical waveform in the normal sample was depicted. The fibrinolysis time was defined as the points where the 50% line crossed the waveform in the curve with transmittance increasing. The decreasing and increasing changes in the curve were considered the coagulation phase and fibrinolysis phase, respectively. **b)** The representative first derivative curve in the normal sample has two peaks in positive and negative values. The first positive peak in the coagulation phase and the second negative peak in the fibrinolysis phase were defined as min1 and |FL-min1|, and considered as the maximum coagulation velocity and the maximum fibrinolysis velocity points, respectively. Six points in the waveform were defined. In the coagulation phase, the start of transmittance decreasing, the 50% transmittance point, and the end of coagulation were defined as coagulation start, max coagulation velocity, and coagulation end. 50% transmittance point is equivalent to maximum velocity. The start of transmittance increasing, 50% transmittance and the end of fibrinolysis were also defined as fibrinolysis start, max fibrinolysis velocity, and fibrinolysis end in the fibrinolysis phase.

## Materials and methods

### Plasma samples

CRYOcheck Pooled Normal Plasma (PNP; Precision BioLogic, Inc. Dartmouth, Canada) was used as the normal plasma sample. Five types of coagulation factor-deficient plasma were used in this study: factor VIII (FVIII), factor IX (FIX), factor X (FX), and factor XI (FXI)-deficient plasma obtained from George King Bio-Medical, Inc. (Overland Park, KS, US). Factor V (FV) deficient plasma was purchased from HYPHEN BioMed (Neuville-sur-Oise, France).

### CFWA method on an automated coagulation analyzer

The method is described in detail elsewhere [6]. Briefly, alteplase (Kyowa Kirin, Tokyo, Japan), a recombinant tPA (r-tPA), was diluted in distilled water. The diluted r-tPA was added to the Thrombocheck $CaCl_2$ solution (Sysmex Corporation, Kobe, Japan), and this procedure was conducted every 2 h to ensure reagent stability [6]. Thrombocheck APTT-SLA (Sysmex Corporation), which includes an ellagic acid activator and synthetic phospholipids, was used as the APTT reagent. In the measurement process, each plasma sample (50 µL) was mixed with APTT reagent (50 µL). After the incubation for 3 min at 37°C, the mixture was also mixed with the $CaCl_2$ solution with r-tPA (50 µL) and the final r-tPA concentration was

0.63 μg/mL in the mixture. The transmittance detection was conducted every 0.1 s at 660 nm wavelength on an automated blood coagulation analyzer CS-2400 (Sysmex Corporation), and clotting time and fibrinolysis time were defined to express the status in each measurement. First, the maximum and minimum values of the transmittance data in the waveform were defined as 0% and 100%, respectively. The 50% transmittance level was determined to be between 0% and 100%. The clotting and fibrinolysis times were defined as the points where 50% of the line crossed the waveform in each phase (Fig 1a). In addition to these two time-related parameters, the first derivative curve indicating coagulation and fibrinolysis velocity is described in the graph according to a previous study [12] (Fig 1b). The absolute maximum values at the coagulation and fibrinolysis phases in the first-derivative curve were considered the maximum coagulation and fibrinolysis velocities and were defined as |min1| and |FL-min1|, respectively. The endogenous fibrinolysis potential (EFP) was calculated from the first derivative graph for each sample [6]. Based on the transmittance data and first derivative graph, six points, including the coagulation start, maximum coagulation velocity, coagulation end, fibrinolysis start, maximum fibrinolysis velocity, and fibrinolysis end, were defined, and the time to reach each point was calculated for each sample (Fig 1). Previously, we demonstrated that the times of maximum coagulation velocity and fibrinolysis velocity were close to the coagulation and fibrinolysis times calculated from the 50% line [6]. We defined and used coagulation and fibrinolysis times as the times required to reach the maximum coagulation and fibrinolysis velocities, respectively.

## Sample preparation and clinical laboratory tests

The CFWA method was conducted on an automated blood coagulation analyzer without stopping, and it was impossible to collect samples during the measurements. To prepare the samples during coagulation and fibrinolysis reactions, the assay was performed outside of the analyzer by mixing plasma sample (50 μL), APTT reagent (50 μL), and $CaCl_2$ solution with r-tPA (50 μL) manually, according to the protocol on the analyzer. Stop solutions containing EDTA 2Na (Dojindo Laboratories, Kumamoto, Japan) and Benzamidine Hydrochloride Monohydrate (Nacalai Tesque Inc., Kyoto, Japan) were prepared at concentrations of 11 mM and 1 M, respectively. The timings at least six points, including coagulation start, maximum coagulation velocity, coagulation end, fibrinolysis start, maximum fibrinolysis velocity, and fibrinolysis end, to add the stop solution were established by the CFWA method on the analyzer for each sample, and 1350 μL of stop solution was added to the mixture prepared manually at each defined time point for stopping the reactions. Samples with stop solution were used for clinical laboratory tests listed below. An Auto LIA FM (Sysmex Corporation) was used to measure FMC as a coagulation marker. For the fibrinolysis markers, FDP (Lias Auto FDP), D-dimer (Lias Auto D-Dimer NEO), and PAP (Lias Auto plasmin-α2 plasmin inhibitor complex; PIC) (Sysmex Corporation) were also employed for measuring fibrin/fibrinogen degradation products (FDP), respectively. All clinical laboratory tests were performed using a CS-2400 analyzer.

## SDS-PAGE and immunoblotting (western blotting)

Each sample for SDS-PAGE and immunoblotting was prepared manually by mixing plasma, APTT reagent, and $CaCl_2$ solution with r-tPA in another aliquot. A Venoject II vacuum blood-collection sample tube (Terumo Venoject, Tokyo, Japan) was used to stop the reaction. A mixture of plasma, APTT reagent, and $CaCl_2$ solution with r-tPA was gently mixed with a granulated powder composed of thrombin, aprotinin, and viper venom in the sample tube. After confirming that the powder had dissolved, the sample was mixed with 2x Laemmli sample buffer (Bio-Rad, Hercules, CA, US) containing 2.1% SDS and incubated at 95°C for 3 min [26]. We added equal volumes to the gel. This procedure was conducted on several samples during the coagulation and fibrinolysis reactions in PNP and on each factor-deficient plasma sample, which were employed as SDS-PAGE. A volume of 4 μL of the sample was applied to 4–20% Mini-Protean TGX gels (Bio-Rad) in each lane, and SDS-PAGE was conducted at 150 V for 1 h. Proteins were transferred to PVDF membranes (Sigma-Aldrich, Tokyo, Japan) using a Bio-Rad mini-transblot apparatus (Bio-Rad) at 50 V for 2 h using the wet method. Polyclonal rabbit anti-human fibrinogen (Dako, Glostrup, Denmark) and monoclonal anti-rabbit peroxidase-linked antibodies (MP

Biomedicals, Aurora, OH, USA) were used as primary and secondary antibodies, respectively. Chemiluminescent signals were detected using enhanced chemiluminescence (Nacalai Tesque, Kyoto, Japan).

## Data analysis

All experiments were performed at least thrice to determine the standard deviation (SD). For FMC, FDP, D-dimer, and PAP, the values relative to the maximum values of PNP were calculated for each parameter, and the percentage values were used for comparison. Microsoft Excel (Microsoft Japan, Tokyo, Japan) and the JMP 17.1 software (SAS Institute Inc., Cary, NC, USA) were used for all data, graphing, and statistical analyses. Statistical significance was set at $P < 0.05$.

'Availability of materials and data` Persistent identifiers: These include a Digital Object Identifier (DOI), accession number, or a link to a permanent record for the data.

## Results

### Coagulation and fibrinolysis parameters in normal and factor-deficient samples

The time required to reach each coagulation and fibrinolysis reaction point was calculated for each sample (Table 1), and the waveforms are shown in Fig 2A. Although there was an interval between the coagulation and fibrinolysis reactions in PNP, the transmittance changes in coagulation phases were very sharp, and there was no clear borderline in the factor-deficient plasma samples. In the FV-deficient plasma sample, this tendency was different from that in the other samples, and the times of maximum coagulation velocity and fibrinolysis initiation were much longer than those in the other samples. The whole reaction of FV required more than 500 sec to complete, so we were unable to observe it to the end. The times of maximum fibrinolysis velocity and fibrinolysis end were not detected because of the analysis timer over error. The times in PNP were the shortest for all coagulation parameters. However, the times related to the fibrinolysis reaction were similar to those of FIX-deficient plasma, and the time of maximum fibrinolysis velocity was longer than that of FIX-deficient plasma. The differences between the coagulation end and fibrinolysis start times for PNP, FVIII, FIX, FX, and FXI were 94.0, 35.4, 21.6, 61.3, and 24.8 s, respectively (Fig 2A). The difference in PNP was the greatest, and the coagulation and fibrinolysis phases were separated in the normal plasma sample. Except for the FV deficiency, the duration from the start to the end of coagulation in normal plasma was 7.6 sec, but in the coagulation factor-deficient plasma group, it was prolonged from 17.5 sec to 74.6 sec. However, the fibrinolytic reaction was 216.8 sec in normal plasma, and there was no difference from other plasma, which was 201.1–241.5 sec (Table 1). These indicated that coagulation factor activity affected the fibrinolysis reactions and related parameters, but the effect was different among the coagulation factors, and the prolongation of time in coagulation parameters was not fully reflected in the fibrinolysis parameters.

**Table 1. The time to react to each coagulation/fibrinolysis reaction point.**

| Point | PNP | FV deficient | FVIII deficient | FIX deficient | FX deficient | FXI deficient |
|---|---|---|---|---|---|---|
| Coagulation start | 24.2±1.1 | N.D. | 75.2±19.8 | 69.4±4.3 | 92.4±1.1 | 83.7±0.1 |
| Maximum coagulation velocity (Coagulation time) | 28.0±1.2 | 351.0±2.8 | 126.7±0.7 | 90.6±3.5 | 101.4±2.2 | 95.8±0.4 |
| Coagulation end (point of 0) | 31.8±1.3 | 391.8±8.6 | 149.8±6.7 | 113.2±4.8 | 109.9±2.9 | 118.9±0.7 |
| Fibrinolysis start (point below 0) | 125.8±4.9 | 410.4±3.2 | 185.2±4.2 | 134.8±1.4 | 171.2±4.4 | 143.7±1.2 |
| Maximum fibrinolysis velocity (Fibrinolysis time) | 207.6±7.1 | N.D. | 277.4±5.6 | 184.6±2.5 | 255.1±3.2 | 281.1±0.3 |
| Fibrinolysis end | 342.6±24.4 | N.D. | 386.3±18.9 | 376.3±4.2 | 397.4±5.9 | 353.1±10.7 |

FIX, factor IX; FV, factor V; FVIII, factor VIII; FX, factor X; FXI, factor XI; N.D., not detected; PNP, pooled normal plasma.

A) Transmittance waveform of CFWA

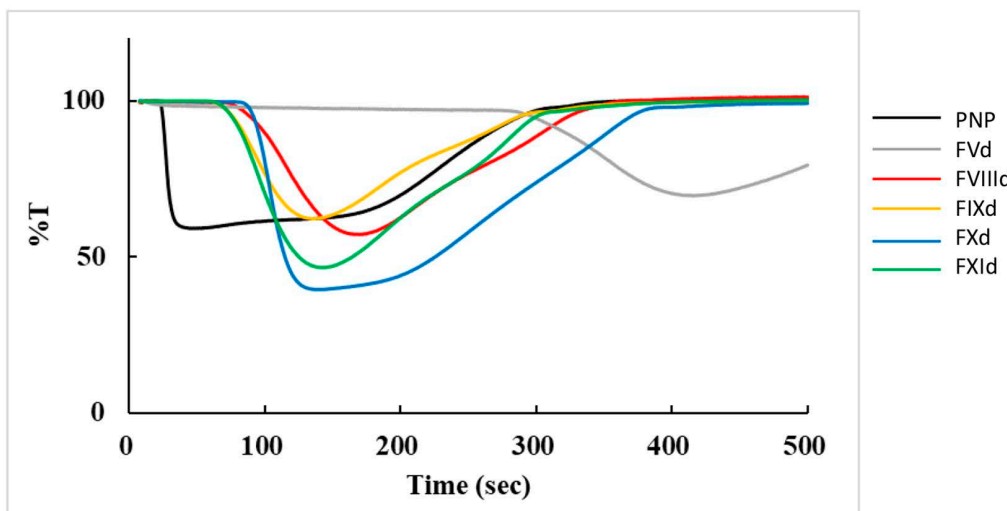

B) Coagulation phase (first derivative curve)

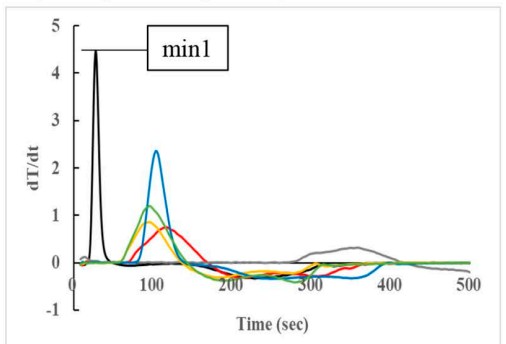

C) Fibrinolysis phase (first derivative curve)

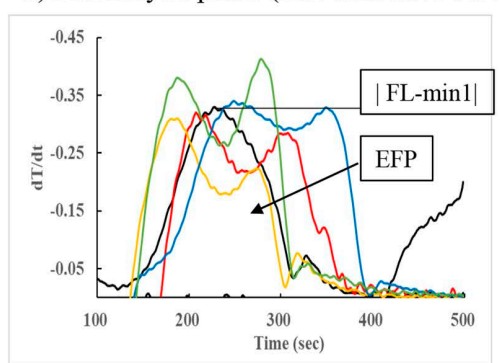

**Fig 2. Clot-fibrinolysis waveform and the analysis in pooled normal plasma and factor-deficient plasma samples. A)** Transmittance waveform of CFWA in pooled normal plasma and five kinds of factor-deficient plasma samples were depicted. The transmittance change was described as %T defined as the changes from the point in time 0. **B)** The first derivative curve was calculated as dT/dt and shown in the coagulation phase, the peak of dT/dt was defined as min1. **C)** The first derivative curve in the fibrinolysis phase was also calculated, and the reversed graph was depicted. The maximum absolute value in the phase and the area were defined as |FL-min1| and EFP.

The transmission light waveform of normal plasma showed that a fibrin clot was formed at a rapid rate in the cuvette. However, in plasma deficient in coagulation factors, the coagulation reaction was disturbed, so the reaction speed was slow, but the speed of the fibrinolytic reaction showed a slope to that of normal plasma.

Fig 2B, C depicts the first derivative waveforms, and Table 2 shows the coagulation and fibrinolysis parameters. The parameter of |min1| in PNP was remarkably 4.46 high for all samples. The |min1| values for all the coagulation factor-deficient plasma groups were low, with factor VIII-deficient plasma showing the lowest value at 0.43. |FL-min1|, there was no clear difference among the five samples. The |FL-min1| of FV-deficient plasma could not be analyzed because the observation time was exceeded. The EFP values of FIX-, FX-, and FXI-deficient plasma samples were higher than those of PNP, and similar fibrinolytic activity levels were observed in factor-deficient plasma samples. In contrast, the EFP of FVIII was lower than that of PNP, in which the fibrinolysis reactions differed among these factor-deficient plasma

**Table 2. Coagulation and fibrinolysis parameters in each sample.**

| Parameters | PNP | FV deficient | FVIII deficient | FIX deficient | FX deficient | FXI deficient |
|---|---|---|---|---|---|---|
| \|min1\| | 4.46±0.36 | 1.22±0.03 | 0.43±0.02 | 0.92±0.06 | 2.36±0.08 | 1.17±0.03 |
| \|FL-min1\| | 0.37±0.01 | N.D. | 0.38±0.02 | 0.31±0.01 | 0.34±0.01 | 0.42±0.01 |
| EFP | 362.3±19.9 | N.D. | 279.2±20.9 | 378.5±2.5 | 565.1±6.5 | 512.3±8.0 |

EFP, endogenous fibrinolysis of potential; FIX, factor IX; |FL-min1|, fibrinolysis min1; FV, factor V; FVIII, factor VIII; FX, factor X; FXI, factor XI; N.D., not detected; PNP, pooled normal plasma.

samples. These reactions showed a decrease in the first derivative curve, with a two-peak pattern in fibrinolysis. The fibrinolysis curve in normal plasma was high and only had one peak.

CFWA, clot-fibrinolysis waveform analysis; CT, clotting time; EFP, endogenous fibrinolysis potential; FIXd, factor IX deficient plasma; FVd, factor V-deficient plasma; FVIIId, factor VIII deficient plasma; FXd, factor XI-deficient plasma; FXId, factor XI-deficient plasma; PNP, pooled normal plasma. (n=3)

### Association between the first derivative curve of clot-fibrinolysis waveform and the related markers

The relationship between the first-derivative curve of CFWA and coagulation and fibrinolysis markers, including FMC, FDP, D-dimer, and PAP, was investigated in PNP-, FV-, FVIII-, FIX-, FX-, and FXI-deficient plasma samples (Fig 3).

The waveforms of the positive and negative values in the first-derivative curve were confirmed as the coagulation and fibrinolysis phases, respectively (Fig 2).

In the PNP sample with a separated phase between the coagulation and fibrinolysis reactions, the fibrinolysis markers FDP, D-dimer, and PAP increased after dT/dt in the first derivative curve returned to 0, and these markers increased as time progressed. This indicates that the fibrinolysis reaction started after the plasma clot was completely formed. In contrast, FMC demonstrated a different tendency; the value increased until the point around the peak of the first derivative curve in the coagulation phase and then decreased to approximately 0%. This value was also increased for other fibrinolysis markers during the fibrinolysis phase, indicating that FMC was created and consumed in the coagulation phase, but was also generated in the fibrinolysis phase. The fibrinolysis markers FDP, D-dimer, and PAP increased before the value of the first derivative curve returned to 0 at the end of the coagulation phase. The FDP and D-dimer values were approximately 40%, which were remarkably lower than those of PNP. Over 50% SD values were recognized for FDP and PAP in the FXI-deficient plasma samples. In the FV-deficient plasma, a clear separation between the coagulation and fibrinolysis phases was not observed, the FMC value did not return to zero, and there was no clear separation between the phases, although the value decreased after the first elevation in the coagulation phase. The change in the balance between coagulation and fibrinolysis was shown to increase in the second half, but it was not possible to observe the entire reaction within 500 sec. The FVIII-deficient plasma sample also demonstrated no clear separation between the phases, and the FMC value did not return to 0. FDP, D-dimer, and PAP levels increased before the first derivative curve demonstrated a negative value in the fibrinolysis phase. The FMC of FVIII-deficient plasma reached its peak at approximately 100 sec after the start of the |min1|, and then decreased from the |FL-min 1| point, showing the highest value during the process of fibrinolysis. There was no difference between coagulation and fibrinolysis, unlike normal plasma. FDP, DD, and PAP increased even before coagulation was complete. In FIX-, FX-, and FXI-deficient plasmas, the tendency was similar. No clear interval between the coagulation and fibrinolysis phases was observed. The first peaks of FMC were observed in the coagulation phase, and an elevation in the fibrinolysis phase was also observed after the decrease. The FMC in FIX deficient plasma increased during coagulation, then decreased once, and then increased again during fibrinolysis. FDP, DD, and PAP showed a pattern of increasing from the latter half of the coagulation reaction, similar to FVIII deficient plasma. FMC values in FX deficient plasma had a peak in the first half and then increased after it returned to the 0 line,

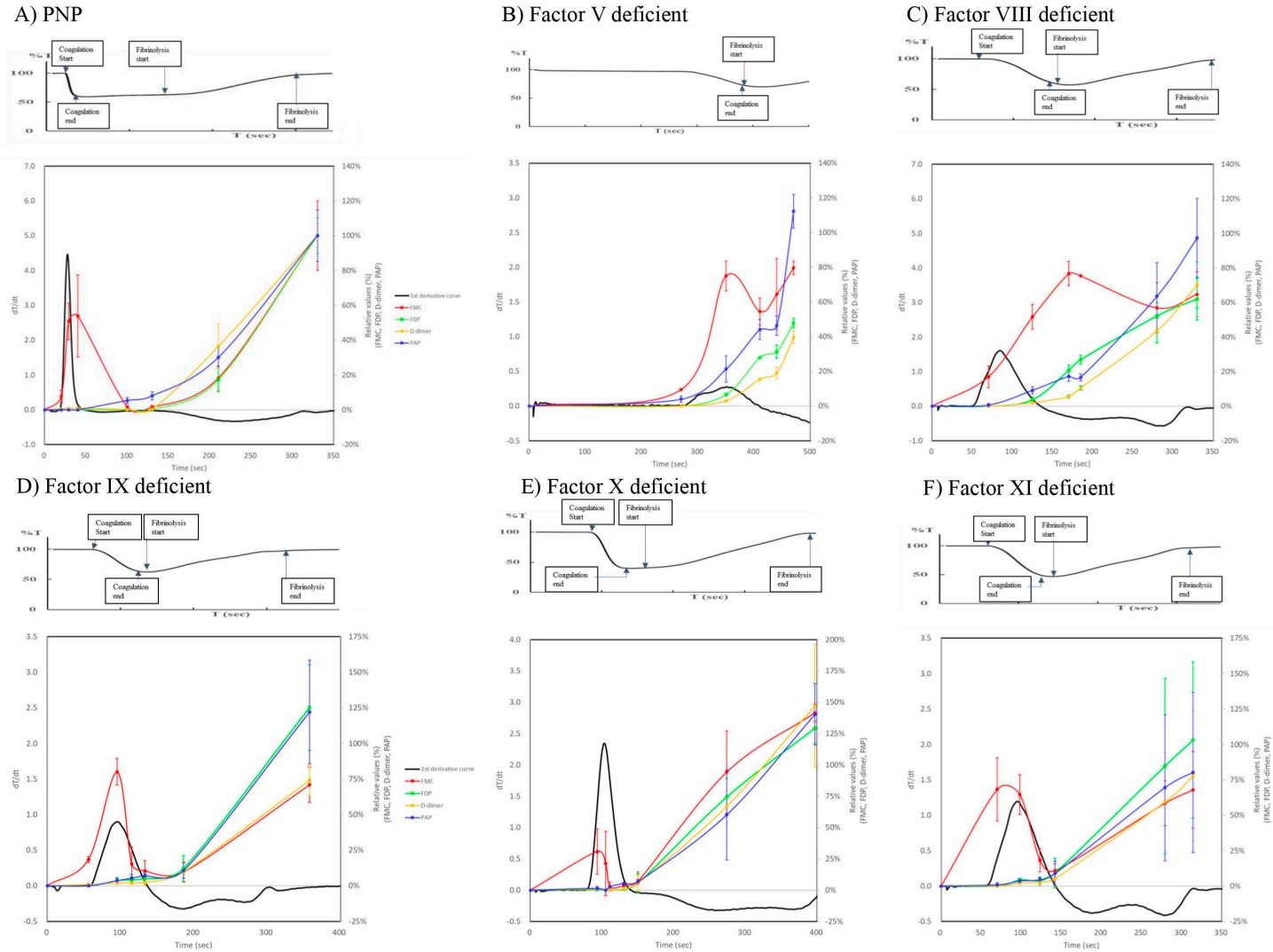

**Fig 3. Association between the first derivative curve of clot-fibrinolysis waveform and the related markers.** The first derivative curve of both the coagulation and fibrinolysis phases in PNP- and FV-, FVIII-, FIX-, FX-, and FXI-deficient plasma samples are depicted with graphs of coagulation and fibrinolysis markers, including FMC, FDP, D-dimer, and PAP. (n = 3) To compare these graph curves, the values of the markers are described relative to the maximum value of the PNP sample. The transmittance waveform is shown in a separate chart at the top of each figure. FMC, fibrin monomer complex; FDP, fibrin./fibrinogen degradation products; PAP, plasmin-$\alpha_2$ plasmin inhibitor complex; PNP, pooled normal plasma.

and the fibrinolytic reaction was higher. FDP, DD, and PAP increased after entering the fibrinolytic phase. FXI deficient plasmas and FIX plasma also showed with FMC detecting two peaks. In FV- and FVIII-deficient plasma samples, we also investigated whether the FMC tendency was recovered by increasing the level of each coagulation factor. The increase in FMC in the fibrinolytic phase indicates that the antibodies used to measure FMC or to produce FMC recognize not only FMC but also small FMC fragments released from fibrinolysis. The graph curves became closer to those of PNP, and two elevations were observed when the coagulation factor concentration was added to each factor-deficient plasma sample (data not shown). Overall, the correlations between reaction times from the starting point and the coagulation and fibrinolysis marker values were statistically significant for FMC ($p = 0.0002$), FDP ($p < 0.0001$), D-dimer ($p < 0.0001$), and PAP ($p < 0.0001$).

## Immunoblotting with anti-fibrinogen polyclonal antibodies

Immunoblotting with anti-fibrinogen polyclonal antibodies was performed on Sodium dodecyl-sulfate polyacrylamide gel electrophoresis (SDS-PAGE) samples (Fig 4). In PNP, only high molecular weight bands, including XDP, DD/E, and X fragments, of over 250 kDa were recognized at the coagulation points. Bands of fibrinogen degradation products, including DD, Y, D, and E fragments, appeared at the fibrinolysis points, and the bands became thicker as time progressed. Overall, the major bands shifted from > 250 kDa to less than 200 kDa. FV-, FVIII-, and FXI-deficient plasma demonstrated D-monomer fragments at the coagulation points, while other fibrinogen degradation products appeared at the fibrinolysis points. There were also some thin bands of fibrinogen degradation products at the coagulation points in the FIX- and FX-deficient plasmas. A similar band shift in PNP was observed in factor-deficient plasma samples, and the band shift began at the coagulation phase in these samples.

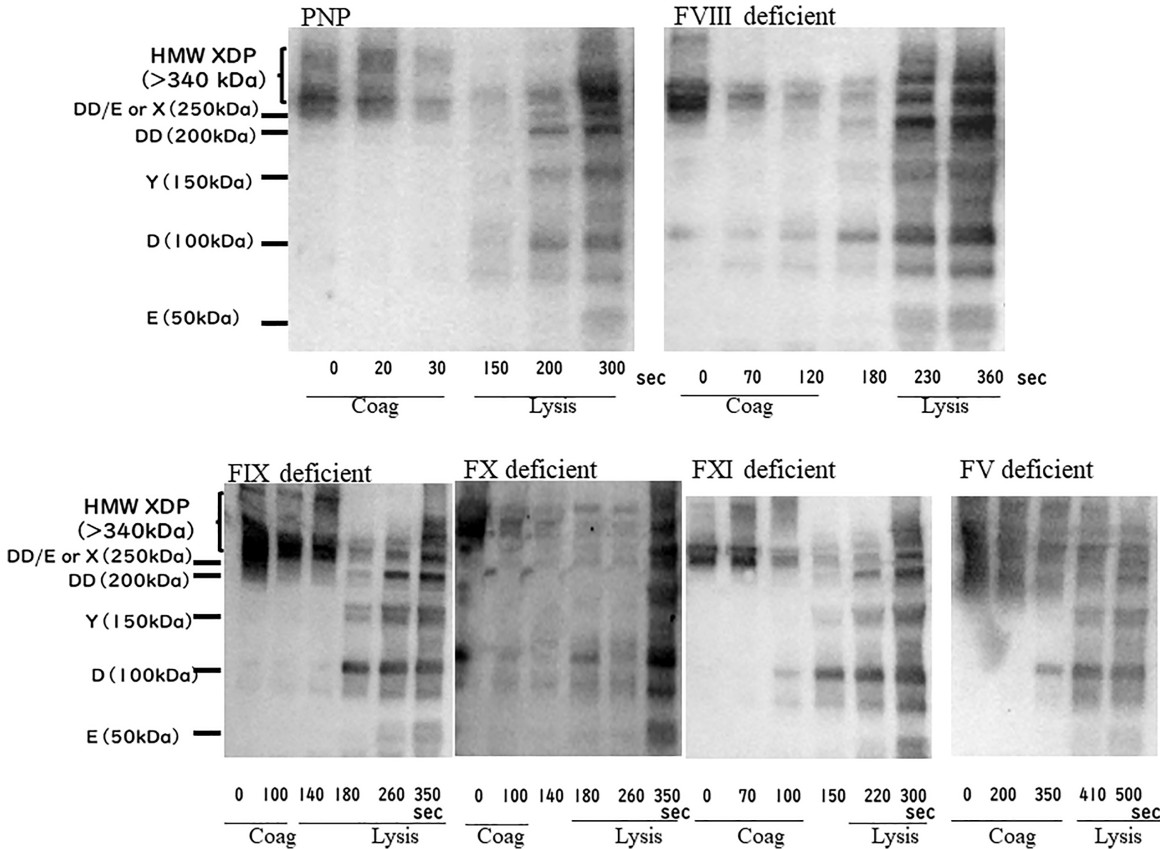

**Fig 4. Immunoblotting in the samples during coagulation and fibrinolysis reactions.** Immunoblotting bands in samples collected at each coagulation and fibrinolysis point are shown. Molecular weight markers and fibrin/fibrinogen degradation products are shown on the left side of the pictures. The coagulation and fibrinolysis phases in each sample are shown based on the results of the transmittance waveform and the first derivative curve. (n = 2) The weight band containing XDP, DD/E and X fragments is 250 kDa or more. The bands of fibrin degradation products (DD/E or X: 250 kDa, DD: 200 kDa, Y: 150 kDa, D: 100 kDa, E: 50 kDa) were shown. Coag, coagulation phase; HMW, high molecular weight; lysis, fibrinolysis phase; PNP, pooled normal plasma.

## Discussion

This study investigated the detailed mechanisms of CFWA and related coagulation and fibrinolysis markers in PNP and five types of factor-deficient plasma samples. In the coagulation and fibrinolysis reactions, these phases were separated in the PNP sample, and the levels of the fibrinolysis markers FDP, D-dimer, and PAP increased after the end of the coagulation phase. However, a clear separation between the coagulation and fibrinolysis phases was not recognized in all factor-deficient plasma samples because the values of the first derivative curve directly changed from positive to negative values without the interval. This indicated that the fibrinolysis reaction started before the end of coagulation in factor-deficient plasma samples. In a previous study, tranexamic acid, an antifibrinolytic agent, prolonged the fibrinolytic reaction of CFWA but did not affect the coagulation phase [6]. It was considered that the lack of coagulation factors affected not only the coagulation phase, but also fibrinolysis, and the clot formation created in these factor-deficient plasma samples would be different from that of PNP.

Samples were prepared by mixing the reaction solution with a stopping solution containing EDTA 2Na to chelate calcium ions and benzamidine hydrochloride monohydrate to inhibit serine protease to measure coagulation and fibrinolysis marker instability. In the stop solution preparation, the plasma: buffer concentration and proportion were established by measuring the residual thrombin and plasmin activities. We confirmed that there were no thrombin and plasmin activities in the chromogenic assay after adding the stop solution. For immunoblotting with anti-fibrinogen antibodies, the samples were immediately mixed with a solution from blood collection tubes and SDS-PAGE sample buffer to denature the proteins. Therefore, we concluded that the residual activities of thrombin, plasmin, and other serine proteases were negligible in these assays.

The FDP, D-dimer, and PAP fibrinolysis markers were used in this study. FDP is generated during fibrinolysis by plasmin and is activated via the proteolytic cleavage of the circulating proenzyme plasminogen, which includes cross-linked fibrin and non-cross-linked fibrin or fibrinogen degradation products [17–20]. In PNP-, FV-, FVIII-, FX-, and FXI-deficient plasma samples, the increasing tendency of FDP was similar to that of the D-dimer, indicating that these fragments were cross-linked fibrins derived from the clot. Only FIX-deficient plasma samples demonstrated different tendencies, and a large discrepancy in the relative values of FDP and D-dimer was observed at the end of fibrinolysis. It was considered that non-cross-linked fibrin and fibrinogen degradation products could be included in the sample and that the clot formation reaction might be different from that of other samples. The PAP, which usually increases during fibrinolysis after plasmin treatment, is inhibited by the $α_2$-PI, elevated over time. The value increased after the end of the coagulation phase in PNP, whereas elevations were observed in the factor-deficient plasma samples during the coagulation phase, indicating that plasmin generation and fibrinolysis began in factor-deficient plasma samples during the coagulation phase. The participation of both α2-antiplasmin and PAI-I performs an important role in the fibrinolytic system. In this study, we did not examine a PAI-I fibrinolytic factors in detail. In the future, it is essential to investigate the relationship with CFWA using assays that reflect α2-antiplasmin and PAI-I. FMC, a small des-AABB fibrin complex, undergoes a two-step reaction in the coagulation and fibrinolysis phases. For FMC measurements, the Auto LIA reagent composed of latex particles with a monoclonal antibody was employed, and the concentration was detected by an immunoturbidimetry assay. The monoclonal antibody used in the reagent was F405, which recognizes the neoepitope exposed through the action of thrombin [21]. It was also reported that this monoclonal antibody detects fibrin monomer degradation products, which are small molecular fragments derived from fibrin monomers via fibrinolytic activity [22,23]. It was considered that the first peak in the coagulation was the fibrin monomer complex and that the molecules were consumed during clot formation. In the fibrinolysis phase, fibrin monomer degradation products generated by fibrinolytic activity appear, and the FMC value increases. The decreasing levels after the first peak in the FV and FVIII factor-deficient plasma samples were smaller than those in the other three factor-deficient plasma samples, suggesting that clot formation may be different in these two factor-deficient plasma samples. FMC tendencies differed among the factor-deficient plasma samples, indicating that individual coagulation factors affected FMC generation. Thus, FMC

may have the reactivity in fibrinolysis phase. It is also useful to measure thrombin-antithrombin complex (TAT) for more classification between coagulation and fibrinolysis phase reaction. Because TAT is considered to increase in only coagulation reaction. The relationship between TAT and FMC was reported in clinical study, which these markers showed similar tendency [24]. Another study to investigate the relationship is planned. High SD values were observed in some samples for the measurements of coagulation and fibrinolysis markers. Measurements were conducted at least three times on manually prepared samples, and the average and SD values were calculated from these data. High SD values were obtained from manually prepared samples. In addition, some results, including the PAP, demonstrated over 50% SD value. The absolute PAP value was < 2.1, whereas the other markers demonstrated higher absolute values. A high SD value derived from the absolute values of each marker was also considered.

Fibrin fibers in normal plasma are fine, and plasmin acts on the surface of fibrin [25]. These reactions were rough in fibrin fibers in plasma deficient in coagulation factors. Lilley L et al. demonstrated that the addition of FVIII increased the density of fibrin fibres, and that the fibrin clot fibres were composed of thinner, more highly branched fibres [26]. In addition, in the present study, by analysing the fibrinolytic reaction, it was considered that in plasma deficient in clotting factors, the density of fibrin fibres decreased, and the fibres were thicker and did not branch as much. Therefore, plasmin was able to reach deep into the fibrin fibers that had formed due to the coagulation disorder. The first peak in the FMC of FV-deficient plasma was due to coagulation, and the subsequent increase was thought to be due to fibrinolysis. In the coagulation factor-deficient plasma group, the coagulation phase min 1 was low, but the FMC was high, and FDP, DD, and PAP increased early. We thought that these reactions may be due to the different quality of fibrin fibers from normal, which makes them more accessible to plasmin.

Immunoblotting using anti-fibrinogen polyclonal antibodies, which detect fibrinogen and fibrin/fibrinogen degradation products, demonstrated that degradation proceeded remarkably over time (Fig 4). The degradation band was not observed at the coagulation points in the PNP sample, indicating that the fibrinolysis reaction started after coagulation. In contrast, degradation bands were observed at the coagulation points in factor-deficient plasma samples, suggesting that fibrinolysis began during the coagulation phase. These results corresponded to the coagulation and fibrinolysis marker measurements. In FVIII- and FX-deficient plasma samples, the bands at the position of the D-monomer [27]. Were recognized at time 0, indicating that the D-monomer fragment existed in the samples before the reaction.

In this study, it was shown that the transmittance changes of CFWA indicated coagulation and fibrinolysis phases and the waveform indicated clot-fibrinolysis reactions. Furthermore, clear differences were observed between the PNP and five factor-deficient plasma samples. The PNP waveform demonstrated a clear separation between the coagulation and fibrinolysis reactions, and the difference between the coagulation and fibrinolysis times was significant. However, a clear separation was not observed in some factor-deficient plasma samples, and the differences were also lower than those of PNP. Thus, the coagulation and fibrinolysis times in CFWA are useful for interpreting patient status.

This study has some limitations. First, all the samples, including PNP- and factor-deficient plasma samples, were purchased. In particular, factor-deficient plasma samples were derived from congenital factor-deficient patients and were considered representative patient plasma samples. Thus, the variety of patients was not reflected, and the different tendencies might have been recognized when more clinical samples were employed. We will have to investigate the results using patient plasma in the future. Second, the tPA concentration used in the assay was much higher than that used under physiological conditions. These results are different from those of the *in vivo* reaction, and the data should be interpreted as *in vitro* results.

## Conclusions

FMC, FDP, and D-dimer were generated from fibrinogen/fibrin in CFWA, indicating that the assay detected coagulation and fibrinolysis reactions by monitoring transmittance. Although fibrinolysis begins after clot formation in normal samples, both reactions occur simultaneously in factor-deficient plasma samples. CFWA is considered a unique tool for identifying

the fibrinolysis status, and the assay would also be useful for investigating the detailed mechanisms of coagulation and fibrinolysis reactions.

## Supporting information

**S1 Fig. Raw images for** Fig 4**. This file contains the raw blot and gel images used in our study.**
(PDF)

**S2 Fig. FMC values in FV- and FVIII-deficient plasma.** This figure presents FMC values in FV- and FVIII-deficient plasma with factor supplementation.
(TIF)

**S3 Fig. Additional immunoblot and Coomassie staining data.** This figure includes supplementary results for fibrinogen and fibrin degradation analysis.
(TIF)

## Acknowledgments

We gratefully acknowledge the technical and interpretive skills of Hiroshi Masutani, Akihiro Nakamura, and Akira Kondo.

## Author contributions

**Conceptualization:** Tomoko Matsumoto.

**Data curation:** Tomoko Matsumoto, Nukumi Tujii, Aya Kouno.

**Formal analysis:** Tomoko Matsumoto, Sho Shinohara, Osamu Kumano.

**Funding acquisition:** Tomoko Matsumoto.

**Investigation:** Tomoko Matsumoto, Nukumi Tujii, Sho Shinohara, Hiroshi Kurono.

**Methodology:** Tomoko Matsumoto, Daiki Shimomura, Aya Kouno, Takeshi Suzuki, Osamu Kumano.

**Project administration:** Tomoko Matsumoto, Daiki Shimomura, Nobuo Arai.

**Resources:** Tomoko Matsumoto, Nobuo Arai.

**Software:** Tomoko Matsumoto, Nobuo Arai, Hiroshi Kurono, Osamu Kumano.

**Supervision:** Tomoko Matsumoto, Osamu Kumano.

**Validation:** Tomoko Matsumoto, Daiki Shimomura, Takeshi Suzuki.

**Visualization:** Tomoko Matsumoto, Osamu Kumano.

**Writing – original draft:** Tomoko Matsumoto, Osamu Kumano, Mikio Kamioka.

**Writing – review & editing:** Tomoko Matsumoto, Osamu Kumano, Mikio Kamioka.

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
