## [Decision Letter · Decision Letter 0]

19 Mar 2025

PONE-D-25-07516Clarification of fibrin generation and degradation reaction processes of clot-fibrinolysis waveform in hemorrhagic disordersPLOS ONE

Dear Dr. Matsumoto,

Thank you for submitting your manuscript to PLOS ONE. After careful consideration, we feel that it has merit but does not fully meet PLOS ONE’s publication criteria as it currently stands. Therefore, we invite you to submit a revised version of the manuscript that addresses the points raised during the review process.

We look forward to receiving your revised manuscript.

Kind regards,

Tomasz W. Kaminski

Academic Editor

PLOS ONE

Journal Requirements:

2**.** Thank you for stating the following financial disclosure:

“This study was supported by JSPS KAKENHI (grant number: JP 21K09089).”

“We gratefully acknowledge the technical and interpretive skills of Hiroshi Masutani, Akihiro Nakamura, and Akira Kondo. This study was supported by JSPS KAKENHI (grant number: JP 21K09089).”

“This study was supported by JSPS KAKENHI (grant number: JP 21K09089).”

“The first author has no conflicts of interest.

T. Suzuki, S. Shinohara, N. Arai, and H. Kurono are employees of Sysmex Corporation. O. Kumano was an employee of the Sysmex Corporation when the study was conducted.”

5. We note that your Data Availability Statement is currently as follows: All relevant data are within the manuscript and its Supporting Information files.

7. We note that you have included the phrase “data not shown” in your manuscript. Unfortunately, this does not meet our data sharing requirements. PLOS does not permit references to inaccessible data. We require that authors provide all relevant data within the paper, Supporting Information files, or in an acceptable, public repository. Please add a citation to support this phrase or upload the data that corresponds with these findings to a stable repository (such as Figshare or Dryad) and provide and URLs, DOIs, or accession numbers that may be used to access these data. Or, if the data are not a core part of the research being presented in your study, we ask that you remove the phrase that refers to these data.

8. Please include your full ethics statement in the ‘Methods’ section of your manuscript file. In your statement, please include the full name of the IRB or ethics committee who approved or waived your study, as well as whether or not you obtained informed written or verbal consent. If consent was waived for your study, please include this information in your statement as well.

9. PLOS ONE now requires that authors provide the original uncropped and unadjusted images underlying all blot or gel results reported in a submission’s figures or Supporting Information files. This policy and the journal’s other requirements for blot/gel reporting and figure preparation are described in detail at https://journals.plos.org/plosone/s/figures#loc-blot-and-gel-reporting-requirements and https://journals.plos.org/plosone/s/figures#loc-preparing-figures-from-image-files. When you submit your revised manuscript, please ensure that your figures adhere fully to these guidelines and provide the original underlying images for all blot or gel data reported in your submission. See the following link for instructions on providing the original image data: https://journals.plos.org/plosone/s/figures#loc-original-images-for-blots-and-gels.  

**Additional Editor Comments:**

Dear Authors,

Thank you for your submission. After careful review, the feedback from the reviewers indicates that the paper is promising and aligns with the scope of PLOS ONE. However, there are several significant issues that need to be addressed before it can be considered for publication.

We recommend that you proceed with revisions to address these concerns. We believe that with the necessary improvements, your manuscript will be in a much stronger position for acceptance.

We look forward to receiving your revised manuscript.

Best regards,

Tomasz W Kaminski

Reviewers' comments:

Reviewer's Responses to Questions

**Comments to the Author**

1. Is the manuscript technically sound, and do the data support the conclusions?

Reviewer #1: Yes

Reviewer #2: Yes

2. Has the statistical analysis been performed appropriately and rigorously? 

Reviewer #1: No

Reviewer #2: N/A

3. Have the authors made all data underlying the findings in their manuscript fully available?

Reviewer #1: Yes

Reviewer #2: Yes

4. Is the manuscript presented in an intelligible fashion and written in standard English?

Reviewer #1: Yes

Reviewer #2: Yes

5. Review Comments to the Author

Reviewer #1: Comments to the Authors

The aim of the present study is to determine whether transmittance changes in CFWA reflect coagulation and fibrinolysis reaction by comparing them with other methods. The authors also compared normal plasma with coagulation factor-deficient plasma samples in CFWA. This study provides a valuable contribution to evaluating CFWA as a clinical application using basic protein assays.

1. The authors used plasma samples provided by company. To detect statistical differences, all experiments were performed three times. However, the authors conducted three experiments using only one sample of each plasma, which suggests that the authors assessed the error within the experimental procedures. The authors should use different lot samples or as mentioned in the limitations, it would be better to use patient plasma.

2. The authors used rabbit anti-fibrinogen antibody provided by MP Biomedicals for immunoblotting. However, I could not find such an antibody on the MP Biomedicals web site. If the antibody was prepared by the authors, the authors should cite a reference that previously used this antibody. If this antibody is being reported for first time, the authors should provide additional data demonstrating its specificity for fibrinogen, including which part of fibrinogen it detects and so on, as this is critical for this study.

3. In the immunoblotting assay, some lanes appear to contain different sample amounts. The authors should include an internal control (e.g., transferrin) or, at the very least, stain the membrane with Coomassie or ponceau S to confirm equal loading.

4. In general, alteplase is used at concentration of approximately 10 -100 nM for in vitro clot lysis assays. Is the amount of r-tPA used in this experiment appropriate?

5. In addition to alpha2 antiplasmin, PAI-1 also play a role in fibrinolysis and should be briefly mentioned in the discussion section.

6. P14, lane 6-7, N.D., note detected, probably not detected?

7. P8, lane 22, FCM should be changed to FMC. The authors should confirm these detail again.

Reviewer #2: In this manuscript, the authors compare the results of their already published assy, the clot-fibrinolysis waveform analysis (CFWA), that simultaneously evaluate coagulation and fibrinolysis in vitro, to the measurements of several specific proteins, relative markers of either coagulation or fibrinolysis.

The paper address some interesting questions but is purely descriptive. Also, the writing could be improved to make it easier to read.

I would have some major comments:

1/ For readers that are not initiated to CWA, it would be important to remind in the introduction that this global assay is performed on plasma and to compare it to whole blood global assays such as thromboelastography and to other plasma CLA (Pieters M, JTH 2019) or thrombin/plasmin generation fluorogenic assays. For example, why use kaolin as a trigger and not thrombin ?

2/ Page 3, line 19: to assess DOAC in vitro effects is different from assessing their clinical efficacy. Please reformulate.

3/ Figures 1A and B are redundant with 1C. Only 1C should be kept.

4/ Figure legends should be separated from the main text and put at the end of the manuscript

5/ Are PIC equivalent to PAP (plasmin-antiplasmin) complexes? If so, the name should be changed.

6/ It is unclear if all laboratory tests performed on CS-2400 analyzer included FDP, D-dimer and PIC. Also, the name of each commercial reagent kit should be added.

7/ The intriguing result from this study is the increase of FMC during the fibrinolysis phase, indicating that FMC are generated during fibrinolysis or that the antibody used to measure FM also recognize small fragments released from FM fibrinolysis... It would have been interesting to compare FMC to thrombin-antithrombin complexes that would really reflect thrombin generation overtime.

8/ Also fibrinolysis markers increase earlier in deficient-plasma compared to normal plasma due to potential less quality fibrin fibers: is there any way to analyse fibrin density on CS-analyzers (from OD value delta at different wavelenghts?)

9/ Results from Western Blot and Figure 4 description should be moved to the Result section. Also the meaning of the different FDP (HMW XDP, DD, Y, D and E fragments) should be explained in the Figure legend.

10/ The number of individual experiments or replicates should be specified at least in each figure/table legend.

6. PLOS authors have the option to publish the peer review history of their article (what does this mean? ). If published, this will include your full peer review and any attached files.

**Do you want your identity to be public for this peer review?** For information about this choice, including consent withdrawal, please see our Privacy Policy .

Reviewer #1: No

Reviewer #2: No

---

## [Author Response · Author response to Decision Letter 1]

1 May 2025

Reviewer's Responses to Questions

Comments to the Author

Reviewer #1: Comments to the Authors

The aim of the present study is to determine whether transmittance changes in CFWA reflect coagulation and fibrinolysis reaction by comparing them with other methods. The authors also compared normal plasma with coagulation factor-deficient plasma samples in CFWA. This study provides a valuable contribution to evaluating CFWA as a clinical application using basic protein assays.

1. The authors used plasma samples provided by company. To detect statistical differences, all experiments were performed three times. However, the authors conducted three experiments using only one sample of each plasma, which suggests that the authors assessed the error within the experimental procedures. The authors should use different lot samples or

We would like to thank you for your comments.

This article used commercially available plasma. We believe that the experiments necessary for investigating the tendency of CFWA were conducted using a single lot. Plasma deficient in factor VIII was tested in three other lots. The reason we selected factor VIII-deficient plasma is because of the severity of bleeding symptoms and the high proportion of patients with hemophilia A in congenital bleeding disorders. The results were similar to those reported in the paper.

However, we agree with you that it would be a good idea to conduct further studies using plasma from other lots or patient plasma. We have added the following to the end of the discussion section.

P.12; lane.34 “We will have to investigate the results using patient plasma in the future.”

2. The authors used rabbit anti-fibrinogen antibody provided by MP Biomedicals for immunoblotting. However, I could not find such an antibody on the MP Biomedicals web site. If the antibody was prepared by the authors, the authors should cite a reference that previously used this antibody. If this antibody is being reported for first time, the authors should provide additional data demonstrating its specificity for fibrinogen, including which part of fibrinogen it detects and so on, as this is critical for this study.

We would like to thank you for your comments.

The reagent was not labeled.

P.6; lane 35 We added the label. This experiment used Polyclonal rabbit anti-human fibrinogen (A0080) from Cite Ab.

3. In the immunoblotting assay, some lanes appear to contain different sample amounts. The authors should include an internal control (e.g., transferrin) or, at the very least, stain the membrane with Coomassie or ponceau S to confirm equal loading.

We would like to thank you for your comments.

In the Immunoblot, we added the same amount of sample to all lanes. We measured the same marker as an internal control in each gel. Furthermore, when examining various deficient plasma, we always performed Immunoblot experiments using pool normal plasma as a control for each measurement.

P.6; lane 29 “We added equal volumes to the gel.”

4. In general, alteplase is used at concentration of approximately 10 -100 nM for in vitro clot lysis assays. Is the amount of r-tPA used in this experiment appropriate?

We would like to thank you for your comments.

As you pointed out, the tPA concentration in CFWA is higher than the therapeutic concentration. CFWA is a method developed by Nogami, K., Matsumoto, T., et al. et al.: A novel simultaneous clot-fibrinolysis waveform analysis for assessing fibrin formation and clot lysis in hemorrhagic disorders. Br J Haematol. 2019 Nov;187(4):518-529. doi: 10.1111/bjh.16111.　 This method reflects coagulation and fibrinolysis processes within 500 seconds, and can be measured quickly using an automatic analyzer.

5. In addition to alpha2 antiplasmin, PAI-1 also play a role in fibrinolysis and should be briefly mentioned in the discussion section.

We would like to thank you for your comments.

The following content has been added to the discussion.

P.11; lane 10 “The participation of both α2-antiplasmin and PAI-I performs an important role in the fibrinolytic system. In this study, we did not examine a PAI-I fibrinolytic factors in detail. In the future, it is essential to investigate the relationship with CFWA using assays that reflect α2-antiplasmin and PAI-I.”

6. P14, lane 6-7, N.D., note detected, probably not detected?

We have corrected the mistake.

7. P8, lane 22, FCM should be changed to FMC. The authors should confirm these detail again.

We have corrected the mistake.

Reviewer #2: In this manuscript, the authors compare the results of their already published assy, the clot-fibrinolysis waveform analysis (CFWA), that simultaneously evaluate coagulation and fibrinolysis in vitro, to the measurements of several specific proteins, relative markers of either coagulation or fibrinolysis.

The paper address some interesting questions but is purely descriptive. Also, the writing could be improved to make it easier to read.

I would have some major comments:

1/ For readers that are not initiated to CWA, it would be important to remind in the introduction that this global assay is performed on plasma and to compare it to whole blood global assays such as thromboelastography and to other plasma CLA (Pieters M, JTH 2019) or thrombin/plasmin generation fluorogenic assays. For example, why use kaolin as a trigger and not thrombin ?

Page 3. line 13: “The thromboelastography can evaluate coagulation and fibrinolysis, but it has low reproducibility (Nagler M,et.al, T�H 2014). The clot lysis time (CLT) with thrombin addition can assess fibrinolysis, but it does not reflect intrinsic or extrinsic coagulation (Pieters M, et.al, JTH 2018). CFWA uses plasma as the measurement sample and an automated coagulation analyser, so results are available within 10 minutes and it has excellent reproducibility, and it can use plasma stored in a freezer. This investigation was performed using a modified APTT reagent to evaluate the bleeding tendency caused by a decrease in clotting factors, including haemophilia, which is an endogenous coagulation disorder.”

We have added the above text and references to the Introduction.

2/ Page 3, line 19: to assess DOAC in vitro effects is different from assessing their clinical efficacy. Please reformulate.

We would like to thank you for your comments.

Page3, line 26; efficacy→in vitro effects

We have made the changes you suggested.

3/ Figures 1A and B are redundant with 1C. Only 1C should be kept.

We would like to thank you for your comments.

We have combined Figure 1 into a single diagram.

4/ Figure legends should be separated from the main text and put at the end of the manuscript

We would like to thank you for your comments.

We have complied with the submission guidelines and prepared this paper, but we have moved the figure legend to the end of the main text as you suggested.

5/ Are PIC equivalent to PAP (plasmin-antiplasmin) complexes? If so, the name should be changed.

We would like to thank you for your comments.

As you pointed out, we changed the PIC to a PAP.

6/ It is unclear if all laboratory tests performed on CS-2400 analyzer included FDP, D-dimer and PIC. Also, the name of each commercial reagent kit should be added.

We would like to thank you for your comments.

We had already written this, but we have made it easier to understand. (Page 6, line 14-19)

7/ The intriguing result from this study is the increase of FMC during the fibrinolysis phase, indicating that FMC are generated during fibrinolysis or that the antibody used to measure FM also recognize small fragments released from FM fibrinolysis... It would have been interesting to compare FMC to thrombin-antithrombin complexes that would really reflect thrombin generation overtime.

Your point is entirely correct. We have added it. The comparison between FMC and TAT is a subject for the next time.

Page 9, line 24 “The increase in FMC in the fibrinolytic phase indicates that the antibodies used to measure FMC or to produce FMC recognize not only FMC but also small FMC fragments released from fibrinolysis.”

We have added the above text to the results.

8/ Also fibrinolysis markers increase earlier in deficient-plasma compared to normal plasma due to potential less quality fibrin fibers: is there any way to analyse fibrin density on CS-analyzers (from OD value delta at different wavelenghts?)

The automated coagulation analyser (CN-6000) reflected the fibrin density by measuring changes in turbidity of transmitted light. Reference 24 stated that when FVIII was added to factor VIII-deficient plasma (0-100 IU/dL), the transmitted light (�Transmittance) increased and the turbidity changes improved.　 This reference cites a report (Lilley L, et al. Res Pract Thromb Haemost. 2017;1:231-241) that found that the addition of FVIII increased the density of fibrin fibres, and that the fibrin clot fibres were composed of thinner, more highly branched fibres.　 We were able to analyse fibrin clots in the absence of clotting factors in detail in our study.

Page 11, line 36 “Lilley L et al. demonstrated that the addition of FVIII increased the density of fibrin fibres, and that the fibrin clot fibres were composed of thinner, more highly branched fibres (25) In addition, in the present study, by analysing the fibrinolytic reaction, it was considered that in plasma deficient in clotting factors, the density of fibrin fibres decreased, and the fibres were thicker and did not branch as much.”

The above text has been added to the discussion, and the references have also been added.

9/ Results from Western Blot and Figure 4 description should be moved to the Result section. Also the meaning of the different FDP (HMW XDP, DD, Y, D and E fragments) should be explained in the Figure legend.

We would like to thank you for your comments.

We have added an explanation of the molecular weight marker and FDP fragment fraction to Figure legend 4.

10/ The number of individual experiments or replicates should be specified at least in each figure/table legend.

We would like to thank you for your comments.

The number of times each item was examined has been added to the figure legend�n= �.

---

## [Decision Letter · Decision Letter 1]

15 May 2025

PONE-D-25-07516R1Clarification of fibrin generation and degradation reaction processes of clot-fibrinolysis waveform in hemorrhagic disordersPLOS ONE

Dear Dr. Matsumoto,

Thank you for submitting your manuscript to PLOS ONE. After careful consideration, we feel that it has merit but does not fully meet PLOS ONE’s publication criteria as it currently stands. Therefore, we invite you to submit a revised version of the manuscript that addresses the points raised during the review process.

We look forward to receiving your revised manuscript.

Kind regards,

Tomasz W. Kaminski

Academic Editor

PLOS ONE

Journal Requirements:

Additional Editor Comments:

Dear Authors,

Thank you for your thorough and thoughtful revision of the manuscript. One reviewer has now recommended a minor revision, with remaining concerns leaning toward clarification rather than substantive change. The second reviewer did not respond to the re-invitation; however, based on my assessment, you have addressed most of their original comments.

However, I believe that one point from the second reviewer’s earlier feedback still requires better attention:

“The intriguing result from this study is the increase of FMC during the fibrinolysis phase, indicating that FMC are generated during fibrinolysis or that the antibody used to measure FM also recognize small fragments released from FM fibrinolysis... It would have been interesting to compare FMC to thrombin-antithrombin complexes that would really reflect thrombin generation over time.”

This comment remains only partially addressed. Strengthening the discussion around this point, particularly regarding the interpretation of FMC levels and the potential value of TAT comparisons, would improve the clarity and impact of your findings.

Given the overall quality of the revision and your responsiveness, I am recommending a decision of Minor Revision.

Best regards,

Tomasz W. Kaminski

Reviewers' comments:

Reviewer's Responses to Questions

**Comments to the Author**

1. If the authors have adequately addressed your comments raised in a previous round of review and you feel that this manuscript is now acceptable for publication, you may indicate that here to bypass the “Comments to the Author” section, enter your conflict of interest statement in the “Confidential to Editor” section, and submit your "Accept" recommendation.

Reviewer #1: (No Response)

2. Is the manuscript technically sound, and do the data support the conclusions?

Reviewer #1: Yes

3. Has the statistical analysis been performed appropriately and rigorously? 

Reviewer #1: I Don't Know

4. Have the authors made all data underlying the findings in their manuscript fully available?

Reviewer #1: Yes

5. Is the manuscript presented in an intelligible fashion and written in standard English?

Reviewer #1: Yes

6. Review Comments to the Author

Reviewer #1: As reviewer #1, I would like to point out some more things to the authors.

In terms of question #2, the polyclonal anti-human fibrinogen antibody (A0080) is provided by Dako, not Cyte ab. It is likely that Cyte ab is the company which provided the data.

And regarding #3, in some immunoblotting results, band intensities appeared to vary between lanes, despite equal loading of samples. To verify that the same amount of protein was loaded in each lane, immunoblotting for serum proteins such as albumin or transferrin should be performed. Alternatively, the transferred PVDF or nitrocellulose membrane should be stained with Ponceau S to assess total protein levels. Another option is to prepare duplicate gels, using one for immunoblotting and staining the other with Coomassie Brilliant Blue to confirm equal loading.

7. PLOS authors have the option to publish the peer review history of their article (what does this mean? ). If published, this will include your full peer review and any attached files.

**Do you want your identity to be public for this peer review?** For information about this choice, including consent withdrawal, please see our Privacy Policy .

Reviewer #1: No

---

## [Author Response · Author response to Decision Letter 2]

11 Jun 2025

Manuscript ID: PONE-D-25-07516R1

Title: Clarification of fibrin generation and degradation reaction processes of clot-fibrinolysis waveform in hemorrhagic disorders

We sincerely thank the Academic Editor and the reviewer for the thoughtful comments and suggestions regarding our revised manuscript. We have carefully addressed all remaining concerns and revised the manuscript accordingly. Below, we provide a point-by-point response to the comments.

Editor’s Comment:

The intriguing result from this study is the increase of FMC during the fibrinolysis phase, indicating that FMC are generated during fibrinolysis or that the antibody used to measure FM also recognize small fragments released from FM fibrinolysis... It would have been interesting to compare FMC to thrombin-antithrombin complexes that would really reflect thrombin generation over time.

This comment remains only partially addressed. Strengthening the discussion around this point, particularly regarding the interpretation of FMC levels and the potential value of TAT comparisons, would improve the clarity and impact of your findings.

Response:

We appreciate this important suggestion and have revised the discussion section to better clarify the interpretation of increased FMC levels during the fibrinolysis phase. Specifically, we now note the following:

Page 11, Lines 29-33�Thus, FMC may have the reactivity in fibrinolysis phase. It is also useful to measure thrombin-antithrombin complex (TAT) for more classification between coagulation and fibrinolysis phase reaction. Because TAT is considered to increase in only coagulation reaction. The relationship between TAT and FMC was reported in clinical study, which these markers showed similar tendency [24]. Another study to investigate the relationship is planned.

We believe these additions have improved the clarity and scientific context of our findings. Although it is difficult to compare FMC tendency with that of TAT due to the lack of several samples, we would like to conduct it as another separated study. Again, thank you for your suggestion.

Reviewer #1 Comments and Author Responses

Comment 1: In terms of question #2, the polyclonal anti-human fibrinogen antibody (A0080) is provided by Dako, not Cyte ab.

Response:

Thank you for pointing this out. We have corrected the manuscript to properly cite the antibody as:

Page 6, Lines 35–36� “Polyclonal rabbit anti-human fibrinogen ( Dako, Glostrup, Denmark)”

Comment 2: In some immunoblotting results, band intensities appeared to vary between lanes, despite equal loading of samples. To verify that the same amount of protein was loaded in each lane, immunoblotting for serum proteins such as albumin or transferrin should be performed. Alternatively, the transferred PVDF or nitrocellulose membrane should be stained with Ponceau S to assess total protein levels. Another option is to prepare duplicate gels, using one for immunoblotting and staining the other with Coomassie Brilliant Blue to confirm equal loading.

Response:

We have taken the reviewer’s advice and performed the suggested verification. Specifically, we used the third approach: preparing duplicate gels, with one used for immunoblotting and the other for Coomassie Brilliant Blue (CBB) staining. This was conducted using FVIII-deficient plasma samples.

Additionally, we measured the protein concentration at each point from 0 to 360 seconds using the BioSpec-nano (Shimazu-biotech) device with absorbance (280 nm) using the same sample as the immunoblot. The detailed methos were as follows.

First, O.D. 280 was detected in BioSpec-nano in each sample.

Second, the protein concentration was estimated from O.D. 280 in the analyzer. Coefficient of molar absorbance of human albumin was used for the calculation tentatively, and the tentative concentration was obtained.

Third, the concentration was compared among the samples. The concentration of Lane 1 (Time 0) was defined as 100%. Each protein concentration value was shown as relative value against sample of Lane 1. Relative value (%) = Concentration of each sample / concentration of Lane 1 (Time 0)

The range of protein concentration was 95.3-107.3% among 6 samples. The variability was within 10%, indicating that the protein concentration was the same level among the samples. Therefore, we suggested that the immunoblotting results reflected the change of fibrinogen/fibrin degradation.

Furthermore, the CBB-stained gel showed comparable band volume values from 0 to 120 seconds, indicating consistent protein loading. After 180 seconds, a decrease in band volume was observed, which we attribute to the degradation of fibrin degradation product (FDP) fractions. These results support the accuracy of our immunoblotting data and the observed time-dependent changes in protein levels. For the detailed data, please find the “Immunoblot and CBB 1,2” files.

We thank the reviewer and editor again for their guidance, which helped us strengthen our manuscript.

Sincerely,

Matsumoto Tomoko., Ph.D.,

Faculty of Health Care, Department of Clinical Laboratory Science, Tenri University,

80-1 Bessho-cho, Tenri, Nara 632-0018, Japan.

Tel.: +81-743-63-7811; Fax.: +81-743-63-6211;

E-mail: t.matsumoto@sta.tenri-u.ac.jp

---

## [Editor Report · Decision Letter 2]

16 Jun 2025

Clarification of fibrin generation and degradation reaction processes of clot-fibrinolysis waveform in hemorrhagic disorders

PONE-D-25-07516R2

Dear Dr. Matsumoto,

We’re pleased to inform you that your manuscript has been judged scientifically suitable for publication and will be formally accepted for publication once it meets all outstanding technical requirements.

Kind regards,

Tomasz W. Kaminski

Academic Editor

PLOS ONE

Additional Editor Comments:

Dear Authors,

Thank you for your thorough revisions. After two rounds of review, I am pleased to confirm that all concerns have been satisfactorily addressed. The manuscript is now in excellent shape, and I have recommended it for acceptance.

Congratulations on your work.
---

## [Editor Report · Acceptance letter]

PONE-D-25-07516R2

PLOS ONE

Dear Dr. Matsumoto,

I'm pleased to inform you that your manuscript has been deemed suitable for publication in PLOS ONE. Congratulations! Your manuscript is now being handed over to our production team.

Kind regards,

on behalf of

Dr. Tomasz W. Kaminski

Academic Editor

PLOS ONE